# Effect of a Multicomponent Intervention on Lifestyle Factors among Brazilian Adolescents from Low Human Development Index Areas: A Cluster-Randomized Controlled Trial

**DOI:** 10.3390/ijerph16020267

**Published:** 2019-01-18

**Authors:** Valter Cordeiro Barbosa Filho, Alexsandra da Silva Bandeira, Giseli Minatto, Jair Gomes Linard, Jaqueline Aragoni da Silva, Rafael Martins da Costa, Sofia Wolker Manta, Soraya Anita Mendes de Sá, Thiago Sousa Matias, Kelly Samara da Silva

**Affiliations:** 1Federal Institute of Education, Science and Technology of Ceara, 63870-000 Boa Viagem, Brazil; 2Post-graduate Program in Collective Health, Ceara State University, 60741-000 Fortaleza, Brazil; jlinard45@gmail.com; 3Research Center for Physical Activity and Health, Federal University of Santa Catarina, 88040-000 Florianopolis, Brazil; alebandeiraufc@gmail.com (A.d.S.B.); gminatto@gmail.com (G.M.); jaqueline.aragoni@outlook.com (J.A.d.S.); rafamc95@yahoo.com.br (R.M.d.C.); sofiawolker@gmail.com (S.W.M.); sorayaanitamds@gmail.com (S.A.M.d.S.); thiagosousamatias@gmail.com (T.S.M.); ksilvajp@gmail.com (K.S.d.S.)

**Keywords:** childhood behaviors, chronic disease, social vulnerability

## Abstract

Promoting healthy lifestyle factors (e.g., physical activity, healthy eating, less screen time) among young people is a relevant and challenging step toward reducing non-communicable diseases. This study aimed to evaluate the effect of a multicomponent intervention on lifestyle factors among adolescents from schools in low Human Development Index (HDI < 0.500) areas. The *Fortaleça sua Saúde* program was conducted with 548 adolescents aged 11–18 years old in the intervention group and 537 in the control group. The four-month intervention included strategies focused on training teachers, new opportunities for physical activity in the school environment, and health education strategies for the school community (including parents). Moderate- to-vigorous physical activity level (≥420 min/week), TV watching and computer use/gaming (<2 h/day), daily consumption of fruit juice, fruit, vegetables, soft drinks, savory foods and sweets, and current alcohol and tobacco use were measured before and after intervention. McNemar’s test and logistic regression (odds ratio [OR] and a 95% confidence interval [95% CI]) were used, considering *p* < 0.05. In the intervention schools, a significant increase occurred in the number of adolescents who met physical activity guidelines (5.3%; 95% CI = 0.8; 9.8) and who reported using computer for <2 h a day (8.6%; 95% CI = 3.8; 13.4) after intervention. No changes were observed in the control schools. At the end of the intervention, adolescents from intervention schools were more likely to practice physical activity at recommended levels (OR = 1.44; 95% CI = 1.00; 2.08) than adolescents from control schools. No significant change was observed for the other lifestyle factors. In conclusion, this multicomponent intervention was effective in promoting physical activity among adolescents from vulnerable areas. However, other lifestyle factors showed no significant change after intervention. This study is registered at Clinicaltrials.gov NCT02439827.

## 1. Introduction

Modifiable unhealthy behaviors, such as tobacco use, excessive alcohol consumption, unhealthy eating, physical inactivity [1] and sedentary behavior [2] are important lifestyle factors which impact the occurrence of non-communicable diseases (NCDs) [3]. An increase in NCDs such as hypertension and type 2 diabetes has been observed worldwide; in particular, these diseases are becoming more prevalent in economically vulnerable areas [4]. For this reason, promoting healthy lifestyles is urgent.

Agendas focused on reducing NCDs emphasize that adolescents’ lifestyles should be a target of the intervention programs [5,6]. Most of the habits are established in this period of life which in turns might influence them later in life [7,8]. The World Health Organization has highlighted actions in schools as one of the most cost effectiveness ways to prevent NCDs in children and adolescents [6], these initiatives are also encouraged by several researchers [9,10,11,12].

However, few studies of school-based lifestyle interventions are multicomponent (i.e., focusing on different lifestyle outcomes) at this point [13,14]. A systematic review found that most interventions target either addiction (drug use, alcohol use, and smoking) or energy balance (nutrition, physical activity, and screen time) [13], but few, if any, have combined these outcomes. Thus studies analyzing synergistic effects from interventions are scarce [13], even though the issue is a priority for public health.

Another central concern is that social and environmental aspects play an important role as determinants of health and lifestyle behaviors [15]. A review of reviews focusing on behavioral interventions concluded that there is an urgent call for studies in a lower socioeconomic context (usually defined as areas with a Human Development Index [HDI] lower than 0.500), as individuals from vulnerable contexts might face more barriers to managing their behaviors and health [16], which contributes directly to health inequities [17,18]. Thus, monitoring efforts and advancement offers is an opportunity to understand the challenges faced by areas that need more attention [19].

Based on these concerns, the present study aimed to evaluate the effect of a multicomponent intervention on lifestyle indicators (physical activity, TV watching and computer use/gaming, healthy and unhealthy eating habits, and tobacco and alcohol use) among adolescents from schools situated in low-HDI (<0.500) municipalities in the northeast region of Brazil.

## 2. Materials and Methods

### 2.1. Study Design and Sample

The present study is a cluster-randomized controlled trial in which schools were the sample selection units. The authorization of the participants’ parents or guardians, through the signing of free and informed consent forms, was a criterion for participation in the research. The National Ethics System (protocol nº 17366313.9.0000.0121) approved this research project. Further methodological details of the study, including a description of the sample size estimate and the flowchart of the sample selection, can be found elsewhere [20,21].

This study was carried out in 2014 in Fortaleza, Northeastern Brazil. All six full-time schools with adolescents enrolled in grades 7–9 and allied to the School Health Program (*Programa Saúde na Escola*, PSE) were considered eligible for the study. The *Programa Saúde na Escola* (Federal Decree No. 6286) aims to provide comprehensive prevention, health promotion and care for children and adolescents who attend schools in the public education system. The schools were randomly allocated to each group, with three in the intervention group and three in the control group.

A total of 1272 adolescents (639 in intervention schools and 633 in control schools) were enrolled in grades 7–9 of eligible schools in the beginning of the study. Of these, 1182 adolescents completed the baseline measures (92.0% and 93.8% of the adolescents in the intervention and control schools, respectively) and 1085 completed follow-up measures (with a response rate of 93.2% and 90.4% in the intervention and control schools, respectively) [21].

### 2.2. Description of the Intervention

The full description of the intervention can be observed in previous publications [20,21]. The intervention was based on socioecological theory [22] and socio-cognitive theory [23], as well as allied with the concept of Health Promoting Schools [24]. The control group schools maintained their regular routine of activities. Strategies regarding the intervention were focused on three components: teacher training, health education, and environmental changes.

The teacher training was given to all teachers from intervention schools at the beginning of the study, and it was focused on two aspects. The first part focused on lifestyle, addressing physical activity and its implications for health, excessive screen time, eating, and prevention of alcohol and tobacco consumption; the second part focused on the relationship of these behaviors to school environment and academic performance. Supportive material was delivered to teachers to assist them in creating both lessons related to the promotion of a healthy lifestyle and strategies proposed by the intervention protocol.

Specific training for physical education teachers was conducted over the same period and duration as that received by teachers from other disciplines. The main impetus was to make physical education classes more active, reaching moderate or vigorous physical activity intensity. Supporting materials were also made available for teachers. In addition to the supporting material, undergraduate physical education adolescents were present throughout the semester to assist the teacher during the classes.

Changes in the school environment were also made to encourage physical activity and decrease sedentary behavior outside classes. Supervised sessions of 10 to 15 min, called “Gymnastics at School”, were performed twice a week. In addition, Games were organized in school places and equipment was made available so that adolescents could play games/sports during free time at school. Strategies involving the school environment also included banners with explanations of the games/sports and health messages involving lifestyle factors.

Strategies focused on health education were enhanced through teacher training, as proposed activities in the classroom and physical education classes involved producing educational material (e.g., posters, newsletters, and health pamphlets) which was then exhibited throughout the school. In addition, pamphlets focusing on physical activity and health, screen time and health, and healthy eating and healthy behaviors were distributed to adolescents, while parents received pamphlets about parental/family involvement in physical activity and screen time. The pamphlets were delivered to a member of the school administration (coordinator or principal) early in the intervention, advising that pamphlets should be distributed at the beginning of school day, during classes, and at parent/teacher meetings at the school.

Considering these strategies, different healthy lifestyle factors were focused on the *Fortaleça sua Saúde* program, as described in Table 1. The primary goals of the *Fortaleça sua Saúde* program were to promote physical activity and reduce screen time; thus, all strategies had content related to these lifestyle factors. Strategies to promote healthy eating comprised teacher training, health lessons in the classroom, and health education messages in the school context. Moreover, substance (alcohol and tobacco) use was also addressed in teacher training, health lessons in the classroom, and the health education pamphlets distributed to adolescents.

### 2.3. Lifestyle Factors and Data Collection

The lifestyle factors examined include physical activity level, screen time (TV watching and computer use/gaming), healthy and unhealthy eating habits, and tobacco and alcohol use. A physical activity list, validated for Brazilian adolescents [21], was used to estimate the weekly time spent in moderate to vigorous physical activity [25]. This 19-item list included team sports (e.g., soccer, basketball, volleyball), individual physical activity/sports (e.g., swimming, athletics), ride-in physical activity (e.g., skateboarding, rollerblading) and popular games (e.g., dodgeball, “forty-forty”). Adolescents reported the frequency and duration of each physical activity that they performed in a typical week. We identified whether adolescents met the World Health Organization’s physical activity guidelines (60 min per day = 420 min per week) [26]. These items showed acceptable reliability (kappa > 0.60) [20].

Screen time during leisure hours was assessed using questions based on the Youth Risk Behavior Survey Questionnaire, which was previously validated for the Brazilian population: daily time watching TV on habitual weekdays (intra-class correlation coefficient/ICC = 0.72) and daily time using computer/video games on habitual weekdays (ICC = 0.80). The cut-off point of less than two hours per day was adopted for each screen time outcome [27].

Eating habits were observed through a questionnaire with three questions related to healthy food indicators (fruits, fruit juice, and vegetables) and three to unhealthy food indicators (soft drinks, sweets, and savory foods) (ICC range = 0.71–0.89) in a typical week [28,29]. We defined healthy consumption as those who consumed each of the healthy food on a daily basis, or who did not consume each of the unhealthy food daily. The frequency of tobacco (ICC = 0.99) and alcohol (ICC = 0.71) consumed in the month preceding the survey was included [30], with healthy consumption defined as those who did not use tobacco or alcohol in the last month.

The control variables were demographic characteristics: gender, age, and socioeconomic status. They were evaluated using a standardized questionnaire, and socioeconomic status was measured using a questionnaire [30] that estimates the purchasing power of families and ranks them according to categories from richest (A1, A2, B1, B2 = A + B) to poorest (C1, C2, D, E = C + D + E) based on the accumulation of material goods, housing conditions, number of working individuals in the household and the education level of the head of household.

The data collection took place in the second semester of 2014, before and after the implementation of the *Fortaleça sua Saúde* program. The questionnaire was administered by evaluators who gave prior instructions and read each question aloud to adolescents in the classroom, who then answered the questionnaire individually. Evaluators were aware which schools were in the intervention and control groups, but they were trained in order to assure standardize measurements regardless of group assignment. Data were computed by scanning using the SPHYNX^®^ software (Sphynx Software Solutions Inc., Washington, DC, USA), with corrections for errors and/or inconsistencies.

### 2.4. Data Analyses

Descriptive statistics were used based on relative frequencies. Proportions of adolescents according to sex, age group, socioeconomic status, and lifestyle factors at baseline were compared between intervention and control groups using the chi-square test. The comparison of the change (follow-up vs. baseline) in proportions of lifestyle factors was performed in each group using McNemar’s test.

Logistic regression was used to identify the odds ratio for the intervention group, when compared with the control group, adopting the healthy categories of the different lifestyle behaviors after follow-up. In the adjusted analysis, models were adjusted for gender, age, socioeconomic status, occupation and outcome variable at baseline, and clustering by school. The level of significance for all analysis was 5% for two-tailed tests using the statistical software SPSS^®^ 23.0 (SPSS IBM Inc., Chicago, IL, USA).

## 3. Results

The sample consisted of 1085 participants. Of these, 51.5% were male and 52.9% were aged 11–13 years-old. A large proportion of adolescents (74.3%) came from lower socioeconomic status households. Three out of ten adolescents (29.7%) met the physical activity guidelines at baseline. The percentage of adolescents who reported watching TV or using a computer for less than two hours a day were 29.1% and 45.5%, respectively. Daily consumption of fruit juices, fruits, and/or vegetables was reported by 20.1%, 18.9%, and 12.4%, respectively. Not consuming soft drinks, savory foods and sweets on a daily basis were reported by 73.6%, 77.9%, and 68.9% adolescents, respectively. A total of 77.4% and 93.9% reported not drinking alcohol and using tobacco in the month previous to the baseline. There was no statistically significant difference between intervention and control adolescents at baseline for lifestyle factors, except for savory foods (*p* = 0.005) and sweets (*p* = 0.017; see Table 2).

In the intervention group, the proportion of adolescents who met physical activity guidelines, who reported watching TV less than two hours a day, and who reported using the computer less than two hours per day increased significantly after follow-up: 5.3% (*p* = 0.016), 6.4% (*p* = 0.004), and 8.6% (*p* < 0.001), respectively. The proportion of adolescents who did not eat sweets daily increased by 8.2% (*p* < 0.001) after intervention. In the control group, significant changes occurred for the proportion of adolescents who reported watching TV for less than two hours per day (increase of 4.7%; *p* = 0.038) and who did not intake sweets daily (increase of 7.4%; *p* = 0.001). No significant changes were observed for other lifestyle factors (consumption of fruit juice, fruits, vegetables, soft drinks, savory foods, or current alcohol and tobacco use) for adolescents from the intervention and control groups (*p* > 0.05). Analyzing the adolescents who adopted healthy lifestyle factors after follow-up (changes at individual level) similar changes occurred for intervention and control adolescents; there was an overlap of the 95% CI for all variables (see Table 3).

After adjusting for confounders, adolescents from the intervention group were more likely (OR = 1.44; 95% CI: 1.00; 2.08; *p* = 0.50) to adopt the physical activity guidelines after follow-up, in comparison to adolescents in the control group. No significant changes were observed for the other lifestyle factors (*p* > 0.05, see Figure 1).

## 4. Discussion

Our results indicated that the *Fortaleça sua Saúde* program had a positive (small) effect on physical activity, but not on other health-related behaviors such as healthy eating and substance use, among adolescents in low-HDI areas in Brazil. This highlights the relevance, but also the difficulty, of finding a comprehensively effective form (i.e., leading to significant changes in different behaviors) of healthy lifestyle intervention in a school setting, particularly in vulnerable areas [15,16].

Positive changes in adolescents among the intervention group were found mainly for outcomes related to an active lifestyle: physical activity level, TV watching and computer use/gaming. The positive changes may be concentrated on physical activity and screen time because most strategies were focused on these lifestyle factors (as shown in Table 1). Indeed, information on the implementation of the program has shown that strategies such as the availability of sports equipment, active lessons in the school curriculum (e.g., English lessons incorporating physical activity) and educational materials (e.g., health risk of high screen time) have contributed to improve attitudes and behaviors toward an active lifestyle [21].

Similar results were observed in other multicomponent interventions involving physical activity among Brazilian adolescents [31,32]. A systematic review of lifestyle interventions focused on obesity prevention among adolescents from low- and middle-income countries also highlighted that multicomponent interventions tend to be more effective in promoting physical activity, and reducing the prevalence of obesity [33] compared to intervention focused on a single component (e.g., a strategy aimed at changing just the school environment).

Effective intervention for reducing screen time was observed with Chinese adolescents. One seven-month school-based peer education program led to a reduction of sedentary behavior by up to 20 min on weekdays [34]. Another study with adolescents from low-income neighborhoods in New York City showed a significant reduction of days per week and hours per day in the use of videogames and watching TV, when compared to the control group [35]. Considering that elevated screen time and low levels of physical activity are frequent in most Brazilian adolescents [36], and that they are associated with NCD risk factors such as high blood glucose and obesity [4,37], our results have shown relevant evidence that multicomponent school-based interventions can be a reasonable strategy for promoting active lifestyles among adolescents who live in a context with higher social vulnerabilities.

Despite being multicomponent, our study did not observe changes in eating consumption, tobacco use, or alcohol consumption. It is important to mention that those variables were not considered primary targets of the intervention, which may lead to a lack of strategy to implement eating or substance use changes. Indeed, strategies on eating consumption and substance use were exclusively focused on knowledge (i.e., without changes on environment and policies such as food availability), which may be insufficient to promote health behavior change [12]. Also, an implementation analysis of the *Fortaleça sua Saúde* program highlighted distinctions between schools in the intervention management (e.g., teachers from all three intervention schools performed health lessons, but almost half of the activities were performed in only one school, while teachers from other schools reported issues such as the absence of education facilities and the lack of planning time). This may reduce the changes on lifestyle outcomes in some contexts and, thus, the total effect of the program [21].

Similar results were observed in the multicomponent CLICK-Obesity Study with Chinese adolescents; from the four eating patterns investigated, it was observed change only in a reduced frequency of red meat consumption (OR = 1.50; 95% CI: 1.15; 1.95). However, the CLICK-Obesity study was feasible and effective in improving changes in awareness of selected risk factors for obesity, such as the importance of consuming fewer calorie-dense foods [38]. Our intervention also included a set of strategies focused on healthy eating education (as shown in Table 1) and may also increase the knowledge of obesogenic eating patterns; however, a behavioral change was not found.

Other evidence from less developed countries shows different results, and changes in eating habits were observed in adolescents from Bangkok, Thailand [39] and Beirut, Lebanon [40]. In both studies, eating was a primary outcome; thus, specified strategies focused on school food policy and school community were implemented.

For tobacco and alcohol use among adolescents no changes were found in the present study. However, a systematic review of lifestyle interventions (focused on tobacco use, alcohol, illicit drug use, risky sexual behavior, and aggressive behavior) showed that, when substance use is a primary target with strategies aimed specifically at decreasing its frequency, multicomponent interventions in school settings tend to be effective in decreasing these behaviors [41]. Our work, therefore, demonstrates the urgency of proposing primary strategies that integrate multicomponent programs aligned with the reduction of tobacco and alcohol use among adolescents from vulnerable areas, since the literature consistently observes a negative association between tobacco and alcohol use and NCDs [42,43,44].

Several limitations of the present study can be addressed. First, the extent to which the intervention actions put against the outcomes were not investigated and can be addressed as a constraint. Also, it is unknown if the synergy among positive outcomes was sufficient to promote changes in eating, tobacco, and alcohol use. It is important to mention that all schools (intervention and control) attempted the national diet program, and it is unknown whether strategies other than ours were applied in order to encourage a healthy eating. The health-related behaviors investigated were self-reported, and the biases that accompany self-reported measures can impact the estimates of the effects (e.g., precision, memory, etc.), especially where food habits are concerned. The duration of the intervention for approximately four months, as well as the lack of maintenance assessments, should be a matter for concern. Other methodological limitations have been further explored in a previous publication [21].

## 5. Conclusions

This multicomponent intervention, which included teacher training, environmental changes, and health education, had a positive effect on physical activity among adolescents from schools in vulnerable areas; relevant changes on screen time among adolescents from intervention schools were also observed. However, no differences were observed in eating consumption or tobacco and alcohol use. Our results showed that the school is a locus for active lifestyle promotion, however designing and implementing interventions that are focused on multiple lifestyle indicators remains a challenge. It is urgent that a global action for healthy lifestyle be implemented in vulnerable contexts, in order to help young people to prevent NCDs and promote health and well-being.

## Figures and Tables

**Figure 1 ijerph-16-00267-f001:**
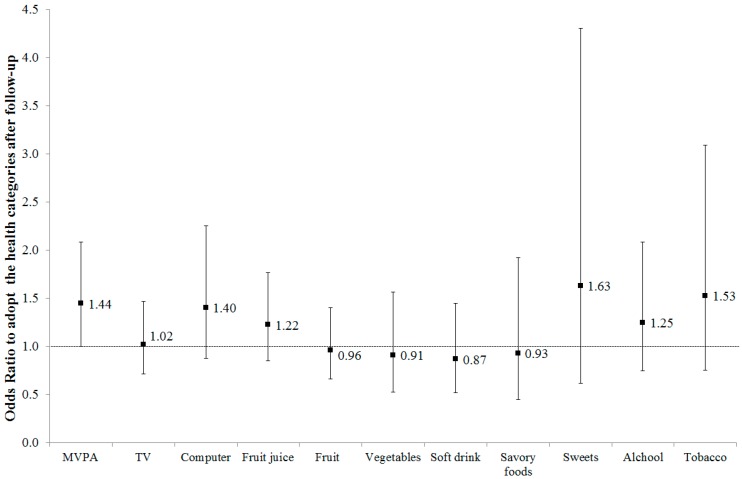
Logistic Regression (Odds Ratio and 95% Confidence Interval) of Intervention Group Adolescents (vs. Control Group Adolescents) Adopting Healthy Lifestyle Factors after Follow-up of the *Fortaleça sua Saúde* Program, 2014. Note: MVPA: moderate-to-vigorous physical activity. Logistic regression adjusted by gender, age, socioeconomic status and lifestyle factor at baseline and clustering by school.

**Table 1 ijerph-16-00267-t001:** Description of the Components and Lifestyle Factors Focused of the *Fortaleça sua Saúde* Program Strategies.

Component Descriptions and Strategies	PA	Screen Time	Healthy Eating	Alcohol and Tobacco Use Prevention
Training and activities in general curriculum and PE classes				
➢ Training with certification focused on health topics and dynamics in the curriculum	✓	✓	✓	✓
➢ Supplemental manual with proposed activities on health topics to be applied in the classroom	✓	✓	✓	✓
➢ Interactive media for teachers to disseminate ideas and implementation of activities in classroom	✓	✓	✓	✓
➢ Exposition of materials (posters, murals) to disseminate health messages in school (integrated with health education)	✓	✓	✓	✓
Active opportunities in the school environment				
➢ Two 10–15 min supervised sessions per week of dynamic activities during free-time in school	✓	✓		
➢ School equipment (balls, rackets, etc.) available to adolescents during free-time in school	✓	✓		
➢ Equipment for games (e.g., mini-courts, “Squash in Health”) with active opportunities and health messages during free-time	✓	✓	✓	
➢ Banners with games rules, material use guidelines and motivational and health messages (integrated with Health Education)	✓	✓	✓	
Health education in school community				
➢ Pamphlets to adolescents in the classroom or schoolyard: (1) PA and health; (2) screen time use and health; (3) eating behaviors	✓	✓	✓	✓
➢ Pamphlets to parents in meetings or visits to schools: (1) PA and family; (2) screen time use and family	✓	✓		

PA: physical activity; PE: Physical Education. ✓ = Lifestyle behaviors that were focused on the intervention strategies of the *Fortaleça sua Saúde* program.

**Table 2 ijerph-16-00267-t002:** Adolescents’ Characteristics at Baseline of the *Fortaleça sua Saúde* Program, 2014.

Variables at Baseline	All (n = 1085)	Intervention (n = 548)	Control (n = 537)	*p*-Value ^b^
Gender		**% (n)**		
Boys	51.5 (559)	51.8 (284)	51.2 (275)	0.903
Girls	48.5 (526)	48.2 (264)	48.8 (262)	
Age groups (years)				
11–13	52.9 (574)	54.2 (297)	51.6 (277)	0.160
14–18	47.1 (574)	45.8 (251)	48.4 (260)	
Socioeconomic status				
A + B (higher status)	25.7 (277)	27.5 (150)	23.8 (127)	0.256
C + D + E (lower status)	74.3 (802)	72.5 (395)	76.2 (407)	
Lifestyle factors				
% adolescents who met PA guideline (420 min/wk MVPA) ^a^	29.7 (322)	28.2 (166)	31.9 (190)	0.159
% adolescents who reported watching TV < 2 h/day	29.1 (316)	28.7 (169)	31.1 (185)	0.367
% adolescents who reported using computer <2 h/day	45.5 (491)	43.7 (257)	46.6 (277)	0.312
% adolescents who consumed fruit juice daily	20.1 (218)	18.7 (110)	22.1 (131)	0.153
% adolescents who consumed fruit daily	18.9 (205)	19.4 (114)	18.2 (108)	0.596
% adolescents who consumed vegetables daily	12.4 (134)	12.4 (73)	12.6 (75)	0.913
% adolescents who did not intake soft drinks daily	73.6 (799)	72.3 (425)	73.1 (434)	0.762
% adolescents who did not intake savory foods daily	77.9 (845)	80.3 (472)	73.4 (436)	**0.005**
% adolescents who did not intake sweets daily	68.9 (747)	71.8 (422)	65.3 (388)	**0.017**
% adolescents who did not intake alcohol in the last month	77.4 (840)	78.7 (463)	75.3 (447)	0.154
% adolescents who did not use tobacco in the last month	93.9 (1019)	93.2 (548)	93.6 (556)	0.779

Abbreviations: Min/wk, minutes per week; PA, physical activity; MVPA, moderate to vigorous physical activity. ^a^ Missing in PA = 2. ^b^
*p*-value were estimated using the Chi-square test, and bold data indicated significant values (*p* < 0.05).

**Table 3 ijerph-16-00267-t003:** Effect of the *Fortaleça sua Saúde* Program on Lifestyle Factors among Brazilian Adolescents from Low-HDI Areas, 2014.

Lifestyle Factors	Prevalence Difference between Follow-Up vs. Baseline (95% CI) ^a^								Prevalence of Adolescents Who Adopt the Outcome after Follow-Up (Individual Level)					
	Intervention (n = 548)				Control (n = 537)				Intervention (n = 548)			Control (n = 537)		
	%	(95% CI)		*p* ^a^	%	(95% CI)		*p* ^a^	%	(95% CI)		%	(95% CI)	
% adolescents who met PA guideline (420 min/wk MVPA)	5.3	0.8	9.8	**0.016**	−3.2	−7.6	1.2	0.141	15.9	13.0	19.2	10.8	8.4	13.7
% adolescents who reported watching TV < 2 h/day	6.4	1.9	10.8	**0.004**	4.7	0.1	9.2	**0.038**	16.4	13.5	19.8	15.8	13.0	19.2
% adolescents who reported using computer < 2 h/day	8.6	3.8	13.4	**<0.001**	3.7	−0.9	8.3	0.098	19.9	16.7	23.5	15.5	12.6	18.8
% adolescents who consumed fruit juice daily	2.0	−2.2	6.2	0.333	−2.4	−6.6	1.8	0.241	12.8	10.2	15.8	10.2	7.9	13.1
% adolescents who consumed fruit daily	0.2	−3.8	4.2	0.926	−0.6	−4.7	3.6	0.783	10.6	8.3	13.5	10.8	8.4	13.7
% adolescents who consumed vegetables daily	0.9	−2.5	4.3	0.579	1.1	−2.4	4.6	0.513	7.8	5.9	10.4	8.4	6.3	11.1
% adolescents who did not intake soft drinks daily	2.9	−1.2	7.1	0.151	2.2	−1.9	6.3	0.265	12.8	10.2	15.8	11.9	9.4	15.0
% adolescents who did not intake savory foods daily	−1.5	−5.6	2.6	0.465	3.2	−1.2	7.5	0.138	10.2	7.9	13.1	13.8	11.1	17.0
% adolescents who did not intake sweets daily	8.2	3.8	12.6	**<0.001**	7.4	3.0	11.9	**0.001**	17.3	14.4	20.7	16.6	13.7	20.0
% adolescents who did not intake alcohol in the last month	−0.7	−4.7	3.2	0.706	0.4	−3.4	4.1	0.837	9.9	7.6	12.7	8.9	6.8	11.7
% adolescents who did not use tobacco in the last month	0.4	−2.2	2.9	0.763	−0.4	−2.6	1.9	0.724	4.2	2.8	6.2	2.8	1.7	4.6

Note: Min/wk: minutes per week; OR: odds ratio; PA: physical activity; MVPA: moderate-to-vigorous physical activity; 95% CI: 95% confidence interval. ^a^
*p*-values obtained using the McNemar’s test for comparisons of proportions, and bold data indicated significant values (*p* < 0.05).

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
