# Peer review of "Effect of a Multicomponent Intervention on Lifestyle Factors among Brazilian Adolescents from Low Human Development Index Areas: A Cluster-Randomized Controlled Trial"

_ijerph, 2019, doi:10.3390/ijerph16020267_

Round 1

Reviewer 1 Report

The paper compares the outcomes of a school-based intervention to improve lifestyles among 11-18-year olds in a multifaceted way among low income students. The intervention is compared to schools which did nothing. The intervention worked with parents and teachers to decrease screen time, increase exercise activity and improve diets particularly increasing vegetable intake. The methodology used was a Difference in Difference method following randomisation of clusters. It is important as tackling Non-Communicable Diseases before they arise is critical in a world where NCD’s are a major cause or morbidity and mortality.

·         Although this is an interesting study after controlling for the demographic variables there was little outcome. This is important and should be reported in a more balanced way. The outcome for a fairly expensive and intensive intervention is very limited (OR 1.44). Screen time is not statistically significant between the intervention and control therefore you can’t report a change. However, this is an important finding. What is clear and has been shown many times is that knowledge alone is not sufficient for behaviour change. The environmental change is critical. This should be discussed as it looks like the intervention mainly addressed knowledge and exercise environment at school.

·         I don’t agree with the discussion that a multicomponent intervention is effective – its not clear what the impact was related to, it is not clear from this evaluation that a single component would not have had the same effect as it was compared to no intervention.

·         It is a pity that gender differences weren’t reported as they are well known to contribute to the determinants.

·         It is not clear why screen time is measured only during weekdays, while exercise is measured over 7 days.

Reviewer 2 Report

The objective of this cluster-randomized controlled trial was to evaluate the effect of a multicomponent intervention on lifestyle 21 factors among students from schools in low Human Development Index (HDI < 0.500) areas. The group case included 548 students aged 11–18 years old in the 23 intervention group and 537 in the control group. The key findings suggests that the multicomponent intervention was effective in promoting physical activity among students from vulnerable areas, but other lifestyle factors showed no significant change after intervention. The authors should consider the following comments:

Point 1. You describe mainly adolescents between 11-18 years - for foreign readers the designation "student" has different meaning. For me better will be using "adolescents" in the title and all manuscript.

Point 2. In the materials and methods section would be appropriate to calculate the sample size in order to better define the power of the study.

Point 3. The good way will be present the sample randomization by using flow chart.

Point 4. What was the reason to adolescents 's division to two groups (table 2)?

Point 5. You should consequently present the results of the study. In table 3 you bolded the main significances, it should be the same way in Table 2 (% students who did not intake savory foods daily : p= 0.005 and % students who did not intake sweets daily: p=0.017).

Point 6. Table 3 – Heading “Adopt these outcomes after follow-up (individual

level), % (95% CI) – the “%” is not necessary.

Point 7. You used the  P-values  to compare to groups : Prevalence difference between follow-up vs. baseline. Why you did not use the P-values in the section: Adopt these outcomes after follow-up (individual level), (95% CI) ?

Point 8. I did not see any of references section in the presented manuscript.

Reviewer 3 Report

A little more discussion of the structural factors would be helpful. In line 57, the word "being" could be dropped.

Round 2

Reviewer 2 Report

Accept in present form